# Assessing the Role of the Perceived Impact of Climate Change on National Adaptation Policy: The Case of Rice Farming in Indonesia

**Mohammad Rondhi** [1,*] [ID]**, Ahmad Fatikhul Khasan** [1,*] [ID]**, Yasuhiro Mori** [2] **and Takumi Kondo** [2]

1   Department of Agribusiness, University of Jember, Jember 68121, Indonesia
2   Research Faculty of Agriculture, Hokkaido University, Sapporo 060-8589, Japan;
    ymouri@agecon.agr.hokudai.ac.jp (Y.M.); kondot@agecon.agr.hokudai.ac.jp (T.K.)
*   Correspondence: rondhi.faperta@unej.ac.id (M.R.); ahmad.fatih@unej.ac.id (A.F.K.)

**Abstract:** Climate change (CC) is one of the primary threats to the agricultural sector in developing countries. Several empirical studies have shown that the implementation of adaptation practices can reduce the adverse effects of CC. The likelihood of farmers performing adaptation practices is mostly influenced by the degree of CC impact that they perceive. Thus, we identified the characteristics of farmers that affect the degree of the CC impact that they perceive. We used data from the Indonesian Rice Farm Household survey consisting of 87,330 farmers. An ordered probit regression model was used to estimate the effect of each variable on the degree of the perceived impact of CC. The results of this study confirm those of previous empirical studies. Several variables that have been identified as having a positive effect on farmer adaptation practices, such as farmer education, land tenure, irrigation infrastructure, cropping system, chemical fertilizer application, access to extension services, and participation in farmer groups, negatively affect the degree of the perceived impact of CC. However, a different result was found in the estimation of the gender variable. We found that female farmers have a higher CC resilience and ability to withstand climatic shocks and risks than male farmers. Female farmers have a more positive perception of future farming conditions than male farmers. We recommend the implementation of a national adaptation policy that use and expand the channel of agricultural extension services to deliver the planned adaptation policy, and prioritizes farmers with insecure land tenure. Additionally, we encourage the increasing of female involvement in the CC adaptation practices and decision-making processes.

**Keywords:** climate change; perceived impact of climate change; climate change adaptation; ordered probit regression; determinants of climate change impact

## 1. Introduction

Climate change (CC) is a global phenomenon that is harming climate-dependent activity such as agricultural production. The negative impacts of CC on agriculture, both for crop and animal production, have been well documented [1–4]. The degree of adverse impacts of CC on farmers is determined by the vulnerability of those farmers to CC. Vulnerability is the propensity or predisposition of a natural or human system to be adversely affected by CC, and encompasses a variety of concepts and elements, including sensitivity or susceptibility and lack of capacity to cope and adapt [5]. Vulnerability, then, is a function of three aspects: Exposure to hazard, sensitivity to damage, and ability to cope. Currently, there are no means to control the occurrence of natural hazards; increasing the system's adaptive capacity can reduce its sensitivity to damage caused by natural hazards [6]. Based on that definition, the implementation of an appropriate adaptation strategy can minimize a farmer's vulnerability.

An appropriate adaptation strategy is required to moderate the adverse effects of CC [7]. There are two types of adaptation: Autonomous and planned adaptation [8]. In the former, farmers independently adapt their farming practices to the observed climatic change. In the latter, the government plans and implements an adaptation policy. Several studies have reported that farmers in developing countries have adjusted their farming practices in response to CC and found that the adaptation has a positive effect on crop yield [9,10]. However, several barriers limit adaptation practices, such as financial barriers (lack of financial resources and/or lack of supporting institutions, whether public or private, to finance adaptation), social and cultural barriers (individuals and group perspectives, values, and beliefs toward CC), and informational and cognitive barriers (individual perceptions, values, and opinions about the risk of CC) [11]. This study focuses on the third barrier, and specifically individual climate risk perception.

A farmer's perception of climate risk is essential because it represents the degree of perceived impact (P-I)—a measure of how a farmer personally feels about the impact of a particular occurrence [12,13]. Past exposure to climate-related disaster increases the degree of P-I, which in turn drives farmers to undertake adaptation actions [14,15]. While some studies stressed the benefits of autonomous adaptation, other studies reported that it ultimately results in unintended maladaptive outcomes, such as increasing the farmer's vulnerability to CC, shifting the vulnerability to other stakeholders or sectors, and decreasing the quality of common pooled resources [16–19]. Thus, assessing a farmer's P-I toward CC is essential in two aspects: First, it provides valuable information about the efforts to encourage autonomous adaptation; second, it provides crucial insight into the effort to avoid maladaptation practices. As most developing countries have a national adaptation policy [20], this study contributes to addressing the question of which farmers should be prioritized and through what channel the content of a policy should be delivered. Figure 1 shows how climate risk perception is related to autonomous adaptation and adaptation outcome.

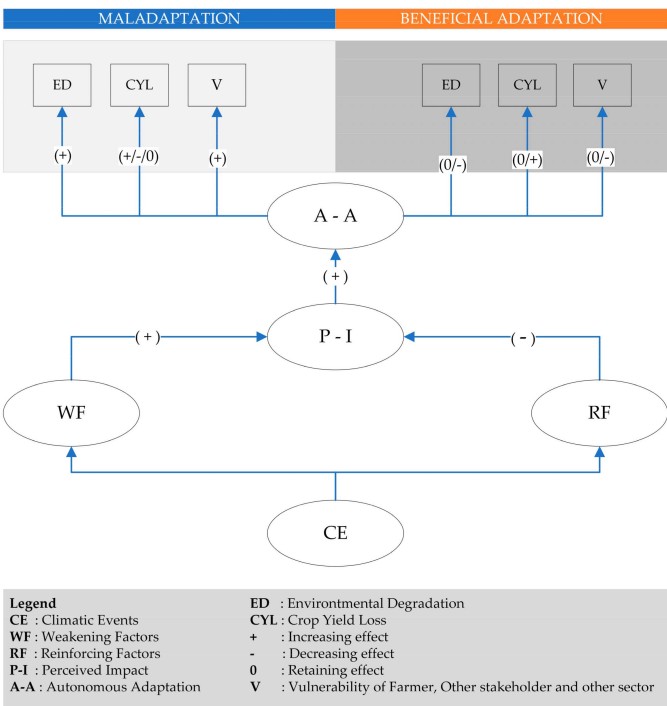

**Figure 1.** The relationship between climate change risk perception, autonomous adaptation, and adaptation outcome.

While CC is a global issue that impacts farmers in both developed and developing countries, it causes more severe damage to those in developing countries [12,21], since the majority of these

farmers are poor and less adaptable [22]. These farmers are often located in less favorable agricultural areas, which increases their vulnerability [23]. The availability of adaptation instruments, whether of a physical nature, such as stress-tolerant crops, or of an institutional nature, such as crop insurance, is limited compared to those in developed countries [24]. Farmers, especially in developing countries, treat CC as a threat to their farming only when they perceive that it significantly decreases crop yield [25,26]. For example, the farmer who experiences crop yield loss due to CC may feel that the impact is not significant because the loss is relatively small compared to their wealth, or due to other reasons, and it is less likely that they will adopt any adaptation strategy. Conversely, a farmer who perceives that the impact of CC is high is more likely to adjust their farming practices to minimize the effect. Based on this reasoning, a farmer's P-I of CC influences their likelihood of adopting an adaptation strategy. The higher the P-I, the more likely the farmer is to adapt, and vice versa. Since the degree of P-I is affected by many factors, including technical, social, economic, or institutional factors [27–29], identifying factors influencing a farmer's P-I of CC is essential. This information will be useful to determine which farmers should be prioritized and targeted.

Extensive studies on the effect of CC on agriculture in developing countries have been conducted. While CC affects almost all aspects of a developing country's economy, agriculture is the most substantially impacted [30,31]. However, few studies have analyzed the effects of CC using nationally representative farm data [32–34]. Nationally representative studies on the impact of CC on agriculture in developing countries often use aggregate data [21,35–38]; other times, individual farm data are employed, and the study is limited to local and community levels [9,39,40]. Whereas the former is useful in providing an overall view of the CC effect and the latter is useful in informing a detailed picture of how a farmer is affected and manages to adapt, it is necessary to conduct a nationally representative study on how farmers are affected by CC. Most of these studies assessed the actual impact of CC on agriculture, which can be different from the farmer perception.

The role of climate risk perception in CC adaptation has received considerable attention. A study in Bangladesh showed that farmer perceptions of CC are mostly aligned with observed meteorological data and are correlated positively with the rate of adopted adaptation practices [15]. Similarly, a study in French coastal populations showed that they perceive the local changes in climate, weather, coral, and beaches, but they only regard it as a problem instead of a danger [41]. In contrast to the result from France, a study on the peri-urban community in Mexico that experiences a risk of drought indicated that the community perceives CC and treats it as a threat because their livelihood as brick producers is severely impacted by climate change [42]. A study on Canadian bivalve aquaculture indicated the importance of stakeholder perceptions of CC in adapting to these changes and further expanding the industry [43]. A cross-country analysis in Europe indicated that the perception of CC is affected by individual-level factors such as gender, age, political orientation, and education, but the size of the effects of each variable varies across countries [44].

The general purpose of this study was to identify the determinants of farmers' P-I of CC. Specifically, we aimed to determine what factors, economic, social, technical, or institutional, affect whether farmers perceive CC as a threat to their farming. The contribution of this study is two-fold. First, this study provides nationally representative information on the farm-level impact of CC in developing countries. While numerous efforts to study the effect of CC on developing countries have been made, few studies have used nationally representative data. Second, we used a P-I measure of CC (whether a farmer thinks they are not, less, or highly affected by CC), which was found to significantly drive adaptation practices. This information is relevant to Indonesia, where a national adaptation policy is being implemented. This study contributes to the growing discussion of climate risk perception.

## 2. Background: Climate Change and Its Impact on Indonesian Agriculture

Indonesia is the largest archipelagic country in the world with over 17,000 islands, with a total land area of 190 million ha. Indonesia had 37 million ha of agricultural land in 2018, and rice fields accounted for 8 million ha, of which 58.13% had irrigation infrastructure [45]. The rice fields are spread

out over the major island in Indonesia, but most of them (3.1 million ha) are located on the island of Java [45]. Rice is a vital food crop in Indonesia along with maize, soybean, green bean, cassava, and sweet potato. The annual production of rice in Indonesia is 81 million tons of dry unhusked rice. Beside food crops, Indonesian agriculture includes horticulture: Vegetables (shallot, chili, cayenne, garlic, and potato), fruits (mango, banana, citrus, durian, pomelo, and mangosteen), and flowers (chrysanthemum, rose, orchid, and tuberose). Additionally, Indonesia produces plantation crops, medical crops, and livestock.

As in many developing countries, agriculture plays a vital role in the Indonesian economy. Its production accounts for 13% of the Indonesian gross domestic product (GDP) [46], and it provides a livelihood for 25 million farm households [45]. Among various crops cultivated in Indonesia, rice is the primary food crop. Of the 25 million farm households, 17 million are rice farmers with an average land possession of 0.6 ha [47]. As a smallholder farmer, the rice farmer is economically more vulnerable to external shocks, such as those due to climatic change. In Indonesia, rice farming has been severely impacted by climate change. The changing rainfall frequency and intensity, the increase in temperature, and the rise in sea level have significantly contributed to declining rice productivity. The frequency of occurrence of extreme events, such as flood and drought, is increasing, which causes crop loss. Increasing temperature causes the proliferation of pests and diseases [48]. Thus, attempts to mitigate and adapt to the risk of CC is required for the resilience of rice farming in Indonesia.

Resilience toward CC is defined as the ability to withstand climatic shocks and risks [49]. Forsyth [50] provides a comprehensive review of the definition of CC resilience. The early definition stated that CC resilience is related mainly to the physical properties, such as infrastructure and ecosystems, and its stability during an occurrence of shocks. The definition is then improved to include not only physical properties but also socio-economic factors such as diverse access to sources of livelihood. Finally, the definition of resilience is brought into a broader context of wider social processes and transformation. For the sake of clarity, this paper defines a farmer's resilience toward CC as their ability to withstand and minimize the adverse impact of CC on their farming.

The study of CC in Indonesia has been extensive and covers a wide range of aspects such as agriculture [48,51–55], natural disasters and management of coastal areas [56–58], the politics of climate change [59–61], and the public perspective on climate change [62,63]. These studies stated that climate change will affect many aspects of the Indonesian economy, and agriculture will be the hardest hit. In response to this, the government created a National Action Plan (NAP) to mitigate the risk of and to adapt to climate change. One of the primary targets of the NAP is to achieve economic resilience by achieving food security. The primary objective in achieving food security is to reduce production loss due to extreme climatic events and CC [64]. The primary strategy to reduce farm production loss is applying a CC-resilient farming system and CC-adaptive farm technology. The government should ensure that the farmer adopts both of these strategy. However, limited government resources and the large number of farmers will limit the adoption rate of this strategy. Thus, it is essential to identify farmers with a high probability to change and adapt, and to target the implementation of the NAP to these farmers.

## 3. Methods

### 3.1. Variable Descriptions

#### 3.1.1. Dependent Variable

The dependent variable is the P-I of climate change on farm yield. A farmer's perception of CC is a subjective measure that represents what impact they think CC has on their farm. Numerous studies have stressed the importance of a farmer's perception toward CC [13,65,66]. This variable was recorded on an ordinal scale to represent the degree of yield loss caused by CC. "No impact" (the farmer perceives no impact of CC on yield), "low impact" (the farmer believes that <50% of yield

loss is due to CC), or "high impact" (the farmer believes that <50% of yield loss is due to CC) were possible variables.

### 3.1.2. Independent Variables

Many factors, including social, economic, technical, institutional, or climatic, may affect the degree of impact of CC on farm production. One factor might reduce the severity of effects because its existence causes farm production to have a stronger resistance toward changing climate so that the farmer can sustain the change without suffering significant yield loss. Alternatively, a factor might reduce the severity of impact because it drives the farmer to take action to limit CC-caused damage. This section briefly reviews the findings of the empirical literature on factors affecting the impact of CC.

#### The Social Factors

We define social factors as the personal characteristics of a farmer. These factors include the age, education, and gender of the farm household head. Several studies have included these factors in identifying how a farmer perceives and adapts to the impact of CC. Several studies have stated that a farmer's age is correlated positively with their resilience against CC. Older farmers have a greater awareness of the impact of CC [67], and their vast farming experience enables them to implement less costly adaptation methods while sustaining a relatively high level of farm productivity [9]. Similarly, the educated farmer copes with CC better than the less educated one, since they can access better information about CC and adaptation technology [9,65,67]. Finally, the issue of gender in CC has received considerable attention because female farmers are more vulnerable to CC, but they have limited access to resources that can be used to adapt [68,69]. Female farmers are less likely to adopt soil conservation methods, cultivate more diverse crops, or plant trees to reduce the effects of CC [65].

#### The Economic Factors

The economic factors are asset-related. We include three variables in this group: Land tenure, landholding, and the source of farm capital. Previous studies have shown that farmers with higher wealth tend to better adapt to CC. Land tenure security is a critical factor for CC adaptation, since it encourages farmers to exert more effort and investment in adaptation practices [70,71]. However, a larger land size increases the cost of adaptation and reduces adaptation practices. Previous studies have stated that farmers with access to credit institutions have a higher probability of adapting to CC [9,65,67]. However, having access to credit institutions does not necessarily mean that a farmer will use borrowed money to obtain a high farm budget. Thus, we use farm capital source instead of mere access to credit institutions.

#### The Technical Factors

We define this category as the technical characteristics of rice farming, which include four variables: Irrigation infrastructure, cropping systems, fertilizer applications, and annual cultivation frequency. Irrigation infrastructure in particular, and agricultural water management in general, play a vital role in mitigating the risk of CC [72–74]. The changing climate alters the frequency of rainfall and affects water availability and crop requirements. Adequate irrigation infrastructure is crucial for the effective distribution of water resources. The farming of mixed species rather than monoculture farming can mitigate the adverse effects of CC [75]. Mixed-species cropping between crops with complementary traits will, with proper management, produce biodiversity and economic advantages in the form of increased productivity. Another significant factor in the technical aspect of farming with respect to CC is fertilizer application [76]. Fertilizer is a primary farm input. However, the excessive use of chemical fertilizer increases the amount of greenhouse gas emissions, which exacerbates CC [77]. The primary challenge of limiting the excessive use of chemical fertilizer is the farmer's perception. A farmer believes that fertilizer application is correlated positively with farm yield. Thus, it is essential to identify how fertilizer application affects how a farmer regards the impact of CC. Similar to the

previous variable, the amount of annual rice cultivation increases the amount of fertilizer usage. Additionally, as the amount of cultivation increases, a farmer will be more exposed to the risk of being impacted by CC. It is essential, then, to identify whether a farmer who cultivates rice more frequently perceives a more severe impact.

The Institutional Factors

Several studies have shown the importance of institutional factors in reducing the impact of CC and encouraging farmers to perform adaptation practices. We include three variables in this category: Participation in farmer groups, access to extension services, and participation in farmer field schools. Conceptually, a farmer group is an important tool for the government to distribute and deliver agricultural policy content to farmers. Participation in a farmer group increases the productivity of the farmer [78] and facilitates members to obtain farm input such as fertilizer and seed [79]. Thus, participation in a farmer group has the potential to increase a farmer's resiliency against climate change. To deliver new information and technology, the government specifically established extension services, and access to these services increases farm performance [80]. In the context of climate change, extension services are the leading channel for the provision of information about climate change and adaptation strategies and technology for farmers. Several studies have indicated that access to extension services increases farmer awareness of CC and their adaptation practices [9,65,67]. Similar to the previous institutions, a farmer field school (FFS) is a government-established service that facilitates the dissemination of new knowledge and skills to farmers. A longitudinal study in East Africa stated that participation in an FFS increases farm productivity by 61% and plays a critical role in reducing poverty [81].

The Climatic Factors

Climate change alters the frequency and intensity of rainfall. In some areas, the intensity of rainfall increases, which causes floods, whereas in mountainous areas, increasing rainfall intensity causes landslides. In other areas, the intensity of rainfall decreases, which causes droughts. All of this CC-caused disaster has a substantial impact on agriculture. Floods and landslides cause severe economic damage including loss of crops, whereas drought reduces the amount of harvested farmland and reduces yield [9,82]. Thus, in this category, we include four types of CC-caused disaster—flood, drought, heavy rain, and other hazards (e.g., landslides)—to determine which disaster the farmer perceives as having the most severe impact on their farming.

*3.2. Data*

We used a nationwide rice farming survey in Indonesia administered by the Indonesian Bureau of Statistics (BPS). The survey was conducted from May 2014 through June 2016 and covered a sample of 87,330 rice farm households (RFHs). The sample selection involved two-stage stratified random sampling. In the first stage, the national BPS office randomly selected census blocks from a total of 844,946 blocks and obtained 8933 sample blocks. Only census blocks with 10 or more RFHs were considered eligible for sample selection. In the second stage, the district BPS office stratified RFHs by land size. Only RFHs with a land size no less than 550 $m^2$ for lowland rice and 100 $m^2$ for upland rice were eligible for sample selection. Figure 2 shows, for each province in Indonesia, the distribution of sample RFHs and the percentage of farmers who experienced the effects of CC.

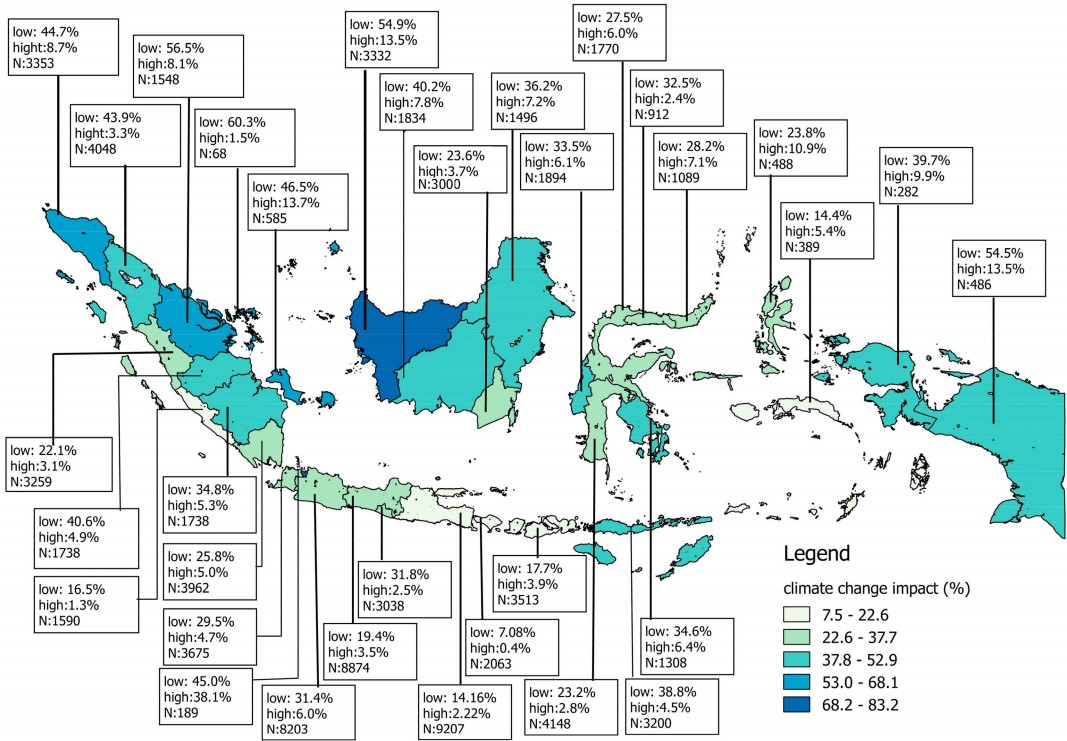

**Figure 2.** The distribution of the sample and the percentage of farmers who experienced effects of climate change (CC) for each province in Indonesia.

Table 1 provides summary statistics, the expected signs of predictors, and the definition of each variable. The outcome variable is the perceived impact severity of CC, which was measured by the question: "Were you impacted by climate change? If yes, how much yield loss was caused by it?" The responses consisted of 4 choices: <25% yield loss, 26–50% yield loss, 51–75% yield loss, and 75% yield loss. The response was then converted into three categories: "no impact" for those who answered 'no' to the first question, "low impact" for those who reported less than 50% yield loss, and "high impact" for those who reported greater than 50% yield loss. The purpose of this further recoding was to clearly distinguish between low and high impact. The original impact category (consisting of 4 groups) yielded a biased estimation between higher-low and lower-low impact and between higher-high and lower-high impact. The data show that 66.2% (57,771 farmers) reported no impact, 29% (25,306 farmers) reported a low impact, and 4.9% (4253 farmers) reported a high impact.

The predictors included 17 variables, which were described in Section 3. The average age of Indonesian rice farmers is 49.5 years, having only elementary schooling (5.81 years, six years of elementary school), and most are male. The majority of farmers (70.7%) cultivate their land and have an average landholding of 0.467 ha. Of rice farmers, 90% financed their farming, leaving only 9% who financed their farm with a loan. The irrigation infrastructure covered 45.3% of the rice farmers in Indonesia, whereas the rest depended on non-technical (46.4%) and rain-fed irrigation (7.9%). Ninety-six percent of farmers only cultivated rice and only 3.9% of farmers cultivated mixed species. Chemical fertilizer is the primary input in rice farming; 91% of farmers apply it, while 8.7% of farmers do not. The average cultivation frequency is twice annually. As with the institutional factors, 52% of farmers participate in a farmer group, 25% have access to extension services, and only 11.2% have participated in a farmer field school. Finally, the climatic variables showed that 7.6% of farmers experienced a flood, 18.9% of farmers experienced a drought, and 5.8% and 1.5% reported experiencing heavy rain and other climate-induced hazards, respectively.

**Table 1.** Summary statistics, the expected signs of predictors, and the definition of variables.

| Variable | Definition | Mean and Frequency | S.D. | Expected Sign |
|---|---|---|---|---|
| Response Variable | | | | |
| Perceived impact of CC | Ordered dummy variable (0 = farmer perceives no impact of CC; 1 = farmer perceives a low impact of CC, production decreases by ≤50%; 2 = farmer perceives a high impact of CC, production decreases by >50%) | 0:57,771 (66.2%) 1:25,306 (29%) 2:4253 (4.9%) | 0.579 | |
| Social Variables | | | | |
| Age | The age of the farm household head (year) | 49.39 years | 11.99 | − |
| Education | The years of formal training of the farm household head (year) | 5.81 years | 4.20 | − |
| Gender | Dummy variable (1 = male, 0 = female) | 1:77,094 (88.3%) 0:10,236 (11.7%) | 0.322 | − |
| Economic Variables | | | | |
| Land tenure | Dummy variable (1 = owned land, 0 = other) | 1:61,784 (70.7%) 0:25,139 (28.8%) | 0.453 | − |
| Landholding | The area of cultivated land (ha) | 0.47 ha | 0.51 | + |
| Capital source | A dummy variable, the source of used farm budget (1 = bank, 0 = other) | 1:79,264 (90.8%) 0:8066 (9.2%) | 0.290 | − |
| Technical Variables | | | | |
| Irrigation | A dummy variable, the type of irrigation infrastructure (1 = technical irrigation infrastructure, 0 = other) | 1:39,530 (45.3%) 0:47,800 (54.7%) | 0.498 | − |
| Cropping system | A dummy variable, the type of cropping system applied (1 = monoculture farming, 0 = mixed-species farming) | 1:83,942 (96.1%) 0:3388 (3.9%) | 0.193 | + |
| Fertilizer | A dummy variable, the application of chemical fertilizer by the farmer (1 = use chemical fertilizer, 0 = does not use chemical fertilizer) | 1:79,744 (91.3%) 0:7586 (8.7%) | 0.282 | + |
| Cultivation frequency | The amount of rice cultivation in a year | 1.64 | 0.696 | + |
| Institutional Variables | | | | |
| Farmer group | Dummy variable, participation in a farmer group (1 = participate, 0 = do not participate) | 1:45,730 (52.4%) 0:41,600 (47.6%) | 0.499 | − |
| Extension services | A dummy variable, access to extension services (1 = having access to extension services, 0 = do not having access to extension services) | 1:21,902 (25.1%) 0:65,428 (74.9%) | 0.433 | − |
| Farmer field school | A dummy variable, access to farmer field school (1 = having access to FFS, 0 = do not having access to FFS) | 1:9762 (11.2%) 0:77,568 (88.2%) | 0.315 | − |
| Climatic Factors | | | | |
| Flood | A dummy variable, experienced a flood (1 = experienced, 0 = did not experience) | 1:6635 (7.6%) 0:80,695 (92.4%) | 0.264 | + |
| Drought | A dummy variable, experienced a drought (1 = experienced, 0 = did not experience) | 1:16,538 (18.9%) 0:70,792 (81.1%) | 0.392 | + |
| Heavy rain | A dummy variable, experienced a heavy rain (1 = experienced, 0 = did not experience) | 1:5069 (5.8%) 0:82,261 (94.2%) | 0.233 | + |
| Other hazards | A dummy variable, experienced other hazards (1 = experienced, 0 = did not experience) | 1:1317 (1.5%) 0:8601 (98.5%) | 0.121 | + |
| Region | A regional dummy variable (1 = Sumatera, 2 = Java, 3 = Bali and Nusa Tenggara, 4 = Kalimantan, 5 = Sulawesi, 6 = Maluku and Papua) | 1:23,379 (27.0%) 2:33,003 (38.2%) 3:8718 (10.1%) 4:9625 (11.1%) 5:10,556 (12.2%) 6:1156 (1.3%) | | |

Note: − and + denote decreasing and increasing effect to farmers' P-I, respectively.

*3.3. Empirical Model*

To estimate the effect of each predictor on the ordinal response variable, we used an ordered probit regression. Ordered probit estimate can be used to estimate how each predictor determines the probability that farmers perceive (whether high or low) an impact of CC on their rice farming. The use of ordered-probit regression to analyze the effect of independent variables on the ordinal response was favored to avoid false alarm (detecting a non-existent effect) and loss of power (failure to detect an effect) problems [83]. Equation (1) specifies the model:

$$y_i^* = \sum_{i=1}^{17} \beta x_i + \varepsilon_i, i = 1, 2, \ldots, N \tag{1}$$

where $y_i^*$ is the response variable that represents the perceived impact of CC, $\beta$ is the parameter to be estimated, $x_i$ is the vector of predictors, $\varepsilon_i$ is the error term, and $N$ is the number of observations.

## 4. Result and Discussion

### 4.1. Estimation Results

Of the 17 predictors analyzed in the ordered probit regression, 13 had a statistically significant effect on the perceived impact of CC, 10 of these had the expected signs, 6 variables had a positive sign, and 7 had a negative sign. However, since the climatic variables logically increase the degree of the P-I, nine variables practically represent farmer characteristics that have a statistically significant effect. The likelihood test ratio and the pseudo $R^2$ indicated that the model is robust.

The estimation results of the social variables revealed that education and gender have a statistically significant effect on P-I, having negative and positive effects, respectively. In the economic category, only land tenure had a statistically significant adverse effect. The technical variables seem to primarily affect the degree of P-I since all variables in this category have a statistically significant effect. Farmers with access to technical irrigation, practicing monoculture farming, and applying chemical fertilizer reported a lower degree of P-I. However, farmers with a higher cultivation frequency seem to have experienced a greater impact of CC. The results of institutional variables suggest that participation in a farmer group and having access to extension services reduce the degree of P-I. However, participation in a farmer's field school is not likely to have a significant effect in terms of decreasing the degree of P-I.

The purpose of incorporating climatic variables in the estimation was to identify which type of climate-related hazard is perceived as causing the most severe damage. The estimation results revealed that all variables in this category have a statistically significant positive effect on the degree of P-I. The obtained coefficients indicated that farmers perceive flood as causing the most damage, followed by drought, heavy rain, and other hazards. The estimation result for each variable is provided in Table 2.

**Table 2.** The estimation results.

| Variable Name | Estimate | Sig. |
| --- | --- | --- |
| **Response Variable** | | |
| Perceived impact of CC | | |
| Threshold *low impact* (1) | −24.133 | 0.000 *** |
| Threshold *high impact* (2) | −19.532 | 0.000 *** |
| **Social Variables** | | |
| Age | −0.001 | 0.513 ns |
| Education | −0.016 | 0.000 *** |
| Gender (Male) | 0.071 | 0.011 ** |
| **Economic Variables** | | |
| Land tenure (Own land) | −0.039 | 0.050 ** |
| Landholding | −0.002 | 0.907 ns |
| Capital source (loan) | 0.030 | 0.340 ns |
| **Technical Variables** | | |
| Irrigation (technical irrigation) | −0.075 | 0.000 *** |
| Cropping system (monoculture) | −0.324 | 0.000 *** |
| Fertilizer (applying fertilizer) | −0.098 | 0.000 *** |
| Cultivation frequency | 0.030 | 0.042 ** |
| **Institutional Variables** | | |
| Farmer group | −0.034 | 0.091 * |
| Extension services | −0.060 | 0.019 ** |
| Farmer field school | −0.001 | 0.980 ns |
| N | 87,330 | |

**Table 2.** *Cont.*

| Variable Name | Estimate | Sig. |
|---|---|---|
| **Climatic Factors** | | |
| Flood | 7.271 | 0.000 *** |
| Drought | 7.086 | 0.000 *** |
| Heavy rain | 6.746 | 0.000 *** |
| Other hazards | 6.682 | 0.000 *** |
| **Regional Variables** | | |
| Sumatera | −0.577 | 0.000 *** |
| Java | −0.394 | 0.000 *** |
| Bali and Nusa Tenggara | −0.549 | 0.000 *** |
| Kalimantan | −0.443 | 0.000 *** |
| Sulawesi | −0.472 | 0.000 *** |
| Maluku and Papua | −0.512 | 0.000 *** |
| **Regression Robustness** | | |
| Likelihoodtest ratio | 135,205.866 | 0.000 *** |
| Pearson goodness of fit | 28,051.967 | 1.000 ns |
| Deviance goodness of fit | 22,696.188 | 1.000 ns |
| Cox and Snell $R^2$ | 0.789 | |
| Nagelkerke $R^2$ | 0.998 | |
| N | 87,330 | |

Note: ***, **, and * denote significant at 1%, 5%, and 10% levels, respectively. ns denotes a statistically not significant effect.

*4.2. Discussion*

The identification of factors affecting the degree of a farmer's P-I of CC was the primary purpose of this study. The estimation results in Table 2 show the effect of each variable. The variables were further classified into weakening and reinforcing factors. A reinforcing factor decreases the degree of P-I, whereas a weakening factor increases it. Figure 3 summarizes the reinforcing and weakening factors.

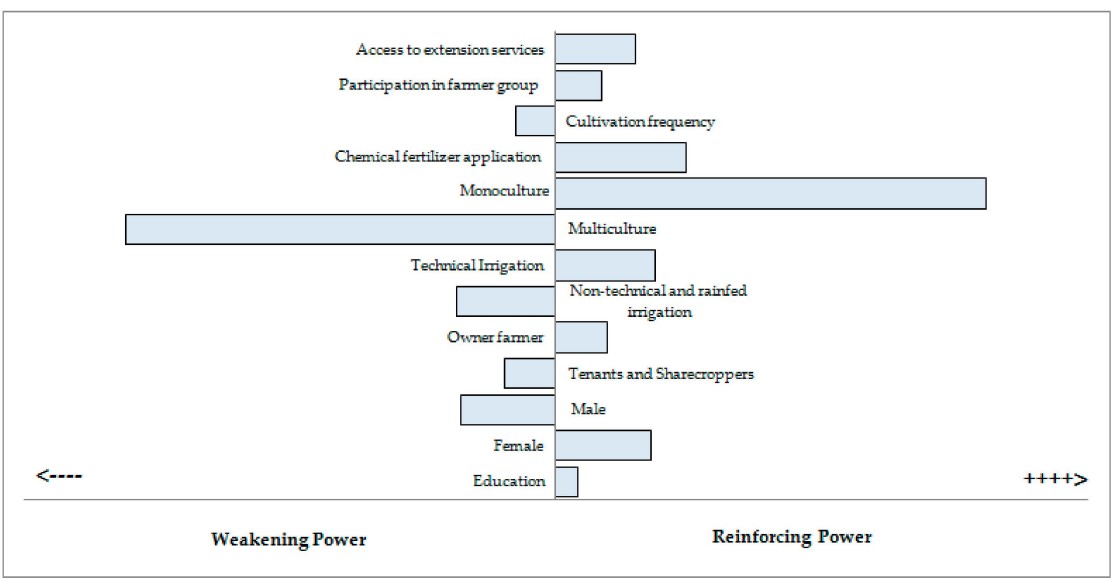

**Figure 3.** The reinforcing and weakening factors of the climate change perceived impact.

4.2.1. Social Variables

The estimation results show that two out of three variables had a statistically significant effect. Education and gender had significantly negative and positive effects, respectively, whereas age did not have any statistically significant effect. The result suggests that farmers with higher levels of education

better adapt to CC and thus perceive a lower degree of CC impact. Many studies have reported the role of education in moderating the adverse effects of CC. A study on farmer use of CC adaptation practices in Pakistan revealed that farmers with higher education are better at adjusting sowing time, using a drought-tolerant crop, and practicing crop-shifting, and these practices have resulted in higher food security [39]. Another study in Pakistan [67] and one in Kenya [84] that assessed farmer awareness of CC also found a positive effect of education on adaptation ability. Similarly, a study in Ethiopia found that education positively affects the probability of farmers adapting to CC via soil conservation and changing planting dates [65]. These findings show that farmers with higher education perceive a lower degree of CC impact because they are better at adapting to and have a higher awareness of CC.

The positive regression coefficient of gender indicates that male farmers perceived a higher degree of impact than female farmers. The gender issue in the CC discussion has gained considerable attention. In some studies, male farmers were reported to be better at adapting to CC. A higher likeliness to undertake adaptation practices was found among male farmers in Ethiopia [65], and a higher awareness of CC was found among male farmers in Kenya [84]. However, in a Pakistan study, female farmers were found to be better at using adaptation practices and had a higher level of food security [39]. A cross-European analysis of individual perceptions of CC revealed that women in most European countries are more aware of CC, but the degree of awareness varied across countries [44]. Also, female representation in the national parliaments is related to the creation of stringent climate change policies and lower $CO_2$ emissions [85]. Furthermore, the theory of socialization stated that female possesses stronger cooperation and carefulness, personal traits that are relevant to the success of CC action, than male [86].

This finding suggests that the gender effect is context- and location-specific. In this study, female farmers perceived a lower degree of impact. Based on the data, female farmers participate more in crop insurance than male farmers. The participation rate of female farmers in crop insurance is 0.255%, whereas that of the male farmer is 0.192%. Women farmers have a more positive perception regarding future farming conditions. Figure 4 shows the distribution of farmer perception of future farming conditions.

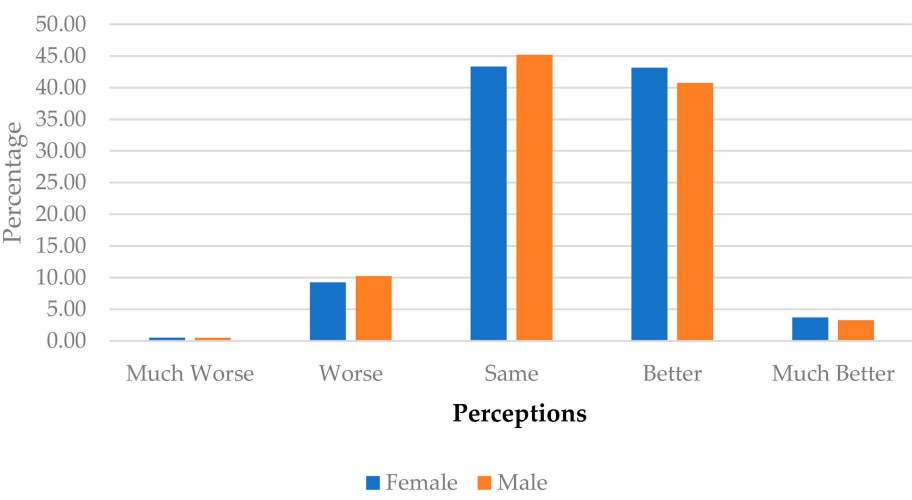

**Figure 4.** Perception of female and male farmers on future farming conditions.

The regression result revealed that age has no statistically significant effect on the degree of P-I. Age has often been associated with farming experience and increases the adoption of adaptation practices [65] and awareness of CC [67,84]. However, another study suggested that younger farmers are more likely to adopt adaptation practices [39]. A cross-European study obtained mixed results in terms of the age effects on individual perception of CC [44]. Thus, the results in this study and of the previous study indicate that, similar to gender, the age effect is location- and context-specific.

### 4.2.2. Economics Factors

The results of this study revealed that land tenure decreases the degree of perceived climate change impact. Landholding and farm capital source did not have statistically significant effects. The effect of land tenure security on the rate of adaptation practices varies. A study in Ghana reported that land ownership increases a farmer's likeliness to undertake adaptation practices [87]. Conversely, another study in Pakistan found that land ownership reduces it [88]. This study suggests that other characteristics, such as education, household size, and resource access, might influence the identified adverse effect of land ownership. Similarly, a study in Pakistan found an adverse effect of land ownership, even after controlling for land size and wealth-related characteristics, but not for education, which was found to have a strong positive effect [39]. These findings suggest that a stronger variable, such as education, might offset the effect of land ownership. The results of this study indicate that land ownership decreases the degree of P-I of CC.

Since we analyzed the P-I of CC on crop yield, the result indicates that farm owners are more willing to adopt yield-enhancing adaptation strategies than farm tenants and sharecroppers. Farm owners have a higher incentive to invest in adaptation practices [89]. A study in Pakistan and Indonesia found that insecure land tenure decreases adaptation practices [90] and makes farmers less cautious in terms of their farming, manifesting, for example, through an absence of soil conservation [91]. The positive effect of secure land tenure on adaptation practices, however, has potential disadvantages when the adaptation yielded unintended maladaptive outcomes. For example, a study on tomato growers in Ghana revealed that the perception of climate variability drives farmer to use more agrochemicals to retain crop production, and these chemicals cause the pollution of nearby water bodies and increase soil acidity above the optimum crop requirements [18]. Similar practices have been identified in rural areas of Indonesia, where watermelon farmers use excessive chemical pesticides to retain production, causing groundwater pollution that damages the quality of water for consumption [92].[1] Ultimately, adaptation becomes maladaptation, which erodes sustainable development and shifts vulnerability to other actors. Thus, it is essential to guide farmer's adaptation practices to avoid unintended maladaptive outcomes.

### 4.2.3. Technical Factors

The probit estimation revealed that each variable in the technical category has a statistically significant effect. The regression coefficient of the irrigation variable indicated that farmers with access to technical irrigation perceive a lower degree of impact than farmers with non-technical and rain-fed irrigation. Irrigation infrastructure is crucial for reducing the adverse effects of CC. The availability of irrigation infrastructure (II) increases the efficiency of the distribution of limited water and decreases crop yield loss due to drought. In some cases, II becomes a drainage facility that reduces crop damage caused by flood. The establishment and improvement of II is a key framing device for CC adaptation in a rice-based country such as Vietnam [94]. In Vietnam, II reduces water availability during the rainy season and increases it during the dry season [95].

Technical irrigation in Indonesia has been developed since the 1970s with the support of the Green Revolution program. The establishment of technical irrigation infrastructure has increased the yields, cropping season, and cropping intensity of rice farming in Indonesia [73]. Conversely, both non-technical and rain fed agriculture rely mainly on rainfall to supply irrigation. Hence, those in such areas are vulnerable to drought occurrence and pay a higher cost of irrigation due to water scarcity. Rice and agricultural productivity in this area are lower compared to areas with technical irrigation, which means that farmers such areas are economically more vulnerable to CC.

The fertilizer variable demonstrated that the application of chemical fertilizer decreases the degree of P-I. In line with the previous discussion, farmers often apply more agrochemicals to retain their crop

---

[1]  The example of data on extensive use of agrochemical in Indonesian farming can be found here [93].

production. A study in Nepal identified that farmers adapt to CC by applying more chemical fertilizer and this increases rice productivity [9]. This finding suggests that autonomous adaptation will likely become maladaptive if a broader perspective is included. This is not surprising as the primary goal of farmers is to increase crop yield or limit crop yield loss, and, in most cases, this is also the priority of agricultural policy. Thus, a further adaptation policy should guide farmers to implement appropriate adaptation practices to reduce unintended maladaptive outcomes.

The results also indicate that monoculture farming decreases the degree of perceived CC impact. Monoculture is often regarded as a threat to biodiversity. Thus, multiculture farming is commonly employed to reduce biodiversity loss in trees farming. However, monoculture in rice farming is less of a threat to biodiversity since its temporal dimension is short [96], and the majority of farmers practice crop rotation. The last variable, cultivation frequency, increases the degree of P-I. A higher cultivation frequency prolongs exposure to CC, which increases the probability of farmers being affected by CC and increases the degree of P-I.

### 4.2.4. Institutional Factors

The estimation results show that access to extension services and participation in a farmer group significantly decrease the degree of P-I, whereas participation in a farmer's field school does not. Extension services are effective information channels used to raise farmer awareness of CC and drive adaptation practices. The importance of extension services as a means of delivering accurate information about CC has been addressed in many studies. A study of farm households in four provinces in Pakistan demonstrated that access to extension services increases how likely a farmer is to adapt to CC [39]. Similarly, a study on rice farmers in the Terai and Hill area of Pakistan indicated that farmers who receive information from extension agents are more likely to adapt [9]. Extension services have been the focus of studies in East Africa [97], Ethiopia [65], and Punjab, Pakistan [67], to enhance a farmer's resilience against CC. These finding indicate that information is crucial in shaping and directing a farmer's behavior with respect to CC. Access to timely and accurate information not only increases a farmer's awareness and adaptation to CC, but might also be used to inform farmers about adaptation practices that yield maladaptive outcomes.

Indonesian farmers receive extension services from several sources, such as state agricultural extension officers (PPL), a state pest-control officer (POPT), an officer from the office of agriculture at the district level (Diperta), and private extension services. Figure 5 shows the number of farmers who receive extension services based on the type of extension service officer. Only 25% of rice farmers in Indonesia have access to these services. The primary extension agent is from the government. The data shows that non-government extension agents vary in type, and these may be from a private corporation (such as a pesticide and fertilizer factory), a non-governmental organization (NGO), or may be associated with peer-farmer extension. Increasing the number and coverage of extension agents is essential for CC adaptation policy in Indonesia, since the majority of farmers (41%) receiving extension services are concentrated in Java, with 21.4% in Sumatera, 8.3% in Bali and Nusa Tenggara, 10.6% in Kalimantan, 14.4% in Sulawesi, and 1.2% and 1.9% in Maluku and Papua, respectively.

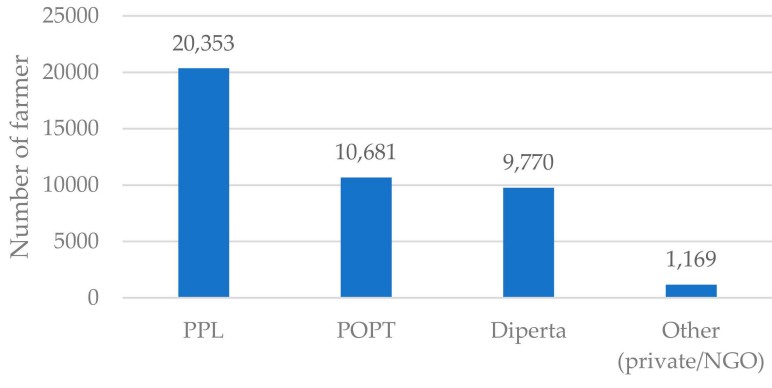

**Figure 5.** The number of farmers who receive extension services based on the type of extension agents.

The government should encourage the establishment of farmer-to-farmer extension services given the potential human resource in Indonesian rice farming. Currently, 82% of the rice farmer population is 20–60 years of age, and 2% of rice farmers have a bachelor degree or higher. Of this educated group (bachelor degree or higher), 87% are aged between 20 and 60 years, with 25% in 20–40 years, and 61% in the 41–60 years old group. Facilitating the establishment of farmer-to-farmer extension provides a strong foundation for sustainable agricultural extension. Farmer-to-farmer extension provides a cost-effective training method for farmers, since trained farmers disseminate their useful knowledge to the non-trained farmer, and the non-trained farmer is eager to adopt the technology used by the trained farmer to increase yield and profit [98]. A strong positive effect of formal education on a farmer's resilience with respect to CC suggests that farmer-to-farmer extension provided by an educated farmer potentially increases the resilience of the recipient.

The estimation results indicate that participation in a farmer group decreases the degree of P-I. Participation in a farmer group is the primary requisite for obtaining farm inputs, extensions, and other programs provided by the government [79]. Several studies have reported the importance of farmer groups in increasing farmer adaptation to CC [65,97], awareness to CC [67], and farm productivity and food security [39].

### 4.2.5. Climatic and Regional Factors

This category focuses on identifying the type of impact that a farmer perceives as most severely damaging their farming. The category contains four variables: Flood, drought, heavy rain, and other hazards (such as landslides). The estimation results show that farmers perceive floods as causing the most severe impact, followed by drought, heavy rain, and other hazards. Flood and drought are the primary consequences of changes in rainfall frequency and intensity. The occurrences of drought and flood are the cause of agricultural yield loss in developing countries and generally affect large areas [99,100]. Figure 6 shows the distribution of farmers who have experienced flood and drought events in Indonesia. The distribution shows that more farmers have experienced droughts than they have floods. Even in provinces with high rainfall intensity, droughts are more frequent than floods. The spatial distribution of affected farmers shows that farmers outside Java are more vulnerable to CC. This information is vital in the implementation of adaptation policies. The government should prioritize the increase in the climate resiliency of farmers outside of Java Island. The increase in resiliency can be achieved by expanding the coverage of agricultural extension services to farmers.

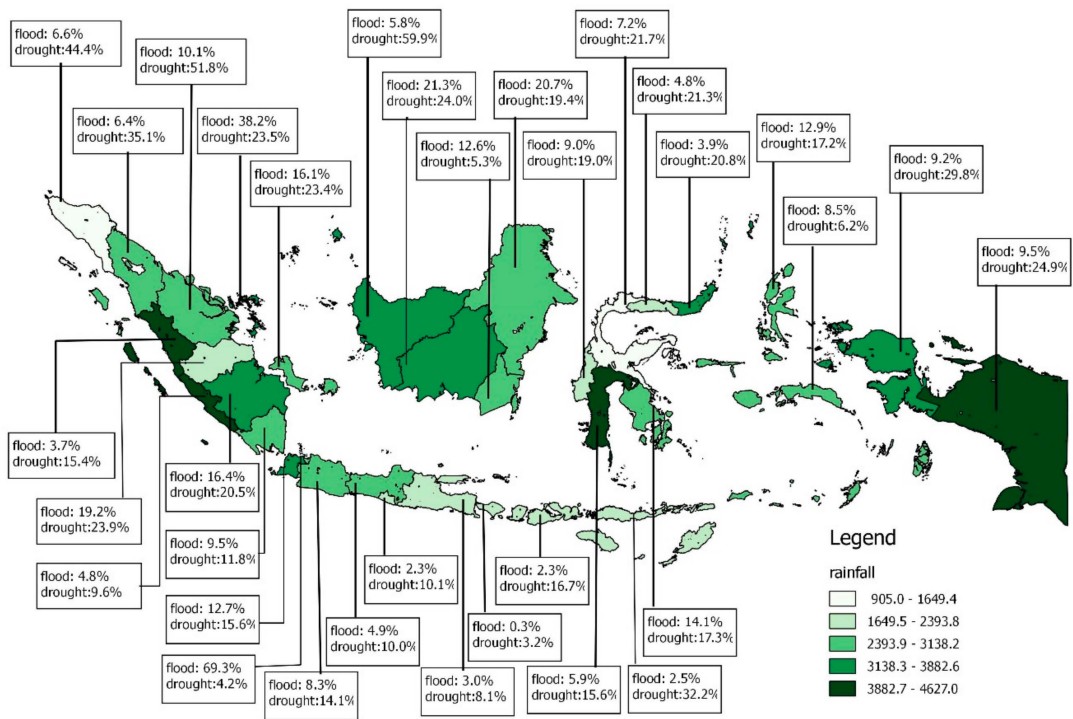

**Figure 6.** The distribution of farmers experiencing flood and drought in each province of Indonesia (the rainfall intensity unit is mm/year).

The regional variables demonstrate regions whose farmers perceive a higher degree of CC. The estimation results indicate that farmers in Sumatera are likely to perceive the impact of CC. Twenty-seven percent of Indonesian rice farmers are in Sumatera, so a large impact of CC on rice farmers in Sumatera is perceived, since the national rice supply significantly decreased. Thirty-eight percent of Indonesian rice farmers in Java perceive a low impact of CC. The agricultural infrastructure in Java is more advanced than other regions in Indonesia. Physical and institutional infrastructure have been established in Java since the 1970s. Thus, in the future, it is critical to emphasize agricultural regions outside Java. The order of regions from the most vulnerable to the most resilient is as follows: Sumatera, Bali and Nusa Tenggara, Maluku and Papua, Sulawesi, and Kalimantan.

### 4.2.6. Policy Implications

The primary purpose of an agricultural CC adaptation policy is to reverse its adverse effects. Adaptation to CC limits agricultural yield loss and improves food security. In some cases, CC is beneficial to agricultural production if proper adaptation practices are implemented [101,102]. Whereas the majority of countries have formulated a specific adaptation strategy to be implemented at the farm level, the remaining challenge is delivering the strategy so that it is adopted by many farmers. The theory of risk perception suggests that individuals who perceive a higher degree of risk would be willing to take any action required to remove the risk [27–29]. In the case of the CC impact on agriculture, farmers will be more willing to adopt adaptation strategy if they individually perceive CC damage. However, this high likeliness to adapt becomes a problem if a farmer's autonomous adaptation creates maladaptive outcomes. Thus, identifying the determinants of a farmer's P-I is crucial in the implementation of adaptation policy. This information can be used to increase the rate of adaptation practices and limit maladaptation.

The results of this study show that the P-I of CC is affected by various factors. Female farmers are shown to be more resilience in managing the impact of CC than male farmers. The theory of socialization suggests that female possesses a stronger personal traits that are relevant to CC action (mitigation and adaptation) than male. Hence, increasing female farmer participation in adaptation

activity and in the CC-related decision-making process is crucial to increase farmer resilience toward CC. Formal education and access to extension services have a strong effect in terms of reducing the degree of P-I, and this strongly agrees with previous studies in many developing countries. Therefore, information is crucial to increasing a farmer's resilience with respect to CC. Thus, we suggest strengthening information about CC as a primary policy. Information about CC can be strengthened in two aspects: First, by increasing the coverage of current extension services and, second, by establishing farmer-to-farmer extension, where educated farmers (those with a bachelor degree or higher) provide education to their peers. Among other factors, land tenure is crucial in adaptation policy. Findings in this study suggest that the security of tenure drives farmers to provide the investment and pay for the running costs of adaptation. Economically, the security of tenure ensures that farmers are incentivized to adopt of adaptation practices.

1.  Policy implications that are suggested based on these findings are as follows: Female farmer are better at managing the CC impact. It is essential to increase female farmer participation in CC action (adaptation and mitigation) and in the CC-related decision making process. Additionally, extension services are currently an effective method of delivering the substance of a policy to the farmer. It is crucial to use current channels of agricultural extension to provide adaptation strategies to farmers and to expand the coverage area of the extension services.

2.  The government should facilitate farmer-to-farmer extension, which can be implemented by identifying key farmers (farmers with a high degree of formal education), providing them with intensive training, and facilitating the dissemination of their expertise. This strategy is feasible because Indonesian rice farmers with a high degree of formal education are aged between 20 and 40 years and have a high potential to be a key farmer in this framework.

3.  Efforts should be made to implement adaptation policies based on a farmer's land tenure and prioritize farmers who cultivate land under a lease or sharecropping contract. The security of tenure affects a farmer's incentive to adopt and conduct adaptation practices. The farmer who cultivates land under a lease or sharecropping contract will put little effort and investment into adaptation practices. Currently, 20% of Indonesian rice farmers are in this category; if they do not practice adaptation strategies, the decrease in rice production will be substantial. In the context of food security, the national rice supply will decrease.

## 5. Conclusions

We attempted to identify factors affecting the degree of P-I of CC on farm yield among rice farmers in Indonesia. The P-I is a subjective measure of the impact of CC on farm yield. This subjective nature of P-I is essential because it indicates how farmers will adapt to CC. The more severe the P-I of CC, the more likely farmers are to adopt adaptation practices. In general, the results seem to support the findings of previous empirical studies and that little difference exists between the actual and perceived impact of CC. Higher education, secure land tenure, the existence of irrigation infrastructure, and access to extension services decrease the degree of the P-I of CC. Previous empirical studies showed that these variables improve how likely a farmer is to adopt CC adaptation practices. Since farmers with a high value for these variables are likely to undertake adaptation practices, we conclude that adaptation limits the adverse effects of CC on farm yield. Hence, the farmer perceives a lower impact of CC.

However, since we measured the perceived impact of CC on farm yield, the existence of maladaptation is suggested. A farmer's primary objective is to retain their yield level, and it is likely that adaptations are primarily aimed to limit yield loss. Consequently, such adaptation potentially creates maladaptive outcomes. In Section 4.2.3, the application of chemical fertilizer was found to decrease the degree of P-I. This suggests that farmers might use excessive chemical fertilizer (inputs) to reduce yield loss. This excessive use will pollute nearby water bodies and decrease soil quality.

Finally, the information obtained from this study is nationally representative and is relevant for the National Adaptation Policy in Indonesia. However, the current study is limited in providing

the details of adaptation practices (the type and form of adaptation) in which farmers engage. Thus, additional study that captures the detailed conditions and practices of Indonesian rice farming is crucial in supporting the information in this study.

Thus, our results provide a basis for future research directions:

1.  As extension services play a vital role in providing information to other farmers, an investigation into the mechanism by which extension services improve the resilience of farmers with respect to CC will provide essential information for improving the efficiency of extension services in delivering timely information to the farmer.
2.  The identification of which type of extension officer and what information contributes most to increasing farmer resilience with respect to CC will yield vital information for increasing overall resilience to CC.
3.  The identification of the type and form of adaptation practices among Indonesian rice farmers and their outcome is crucial for identifying whether a farmer's practices will lead to appropriate adaptation or maladaptive outcomes.

**Author Contributions:** Conceptualization: M.R., A.K., Y.M. and T.K.; formal analysis: M.R., A.K. and Y.M.; funding acquisition: T.K.; investigation: Y.M.; resources: Y.M. and T.K.; software: M.R. and A.K.; supervision: M.R. and T.K.; visualization: M.R. and A.K.; writing—original draft: M.R. and A.K.; writing—review & editing: M.R.

**Funding:** This research was funded by the Japan Society for the Promotion of Science, Grant Number: 18K05839.

**Conflicts of Interest:** The authors declare no conflict of interest.

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
