# Peer review of "Assessing the Role of the Perceived Impact of Climate Change on National Adaptation Policy: The Case of Rice Farming in Indonesia"

_land, doi:10.3390/land8050081_

Round 1

Reviewer 1 Report

Comments:

1.      This paper addressed an interesting topic; it aimed to explore the characteristics of the farmer affecting the degree of farmer’s perceived-impact of climate change.

2.      The title is very long; I suggest replacing it with the following: “Assessing the role of perceived-impact of climate change in national adaptation policy: The case of Indonesia”

3.      The abstract should be improved by adding 1-2 warm-ups sentences as an Introduction to the topic.

4.      Line 14 “We use data from the Indonesian Rice Farm…” the authors should avoid the use of personal pronouns within the body of the manuscript (e.g. "this paper investigates..." is correct; "I/we investigate..." is incorrect). Please revise the whole manuscript.

5.      The keywords should be improved as the term “climate change” has been mentioned many times (e.g. perceived-impact of climate change; climate change adaptation; determinants of climate change impact).

6.      Line 34 “An appropriate adaptation strategy is…” the authors discussing the adaptation strategies too soon. Here the authors should discuss the concept of vulnerability to climate change including its definition and other aspects in one paragraph; afterwards, they can introduce the aspect of adaptation and responses to negative consequences of climate change.

7.      Line 76 “Previous studies on the effect of CC on agriculture…”; the authors should cite these studies.

8.      The main contribution of the current study compared to similar studies should be highlighted; it could be done in the Introduction section in one paragraph.

9.      The authors should add a few relevant/recent studies along with their approaches and outcomes and indicate the main contribution of the current study by comparing it with previous ones.

10.  Overall, the authors need to carefully revise the language issues as at several places in the text as they wrongly used adjectives and pronouns like “which”. Similarly, authors also need to look for the use of verb and tenses for third-person, singular and plural nouns.

11.  Line 210 “The survey was conducted on May 2014 through June 2016…” what took you so long to publish the results?

12.  The quality of Figure 1 should be enhanced in the Methods section.

13.  Line 221 “Table 1 provides the summary statistics…”; the authors should avoid using very short paragraph and they should either extend the paragraph or merge it with the rest of the text.

14.  The discussion of the results according and compared to existent literature is missing. The authors should outline how the main findings are in line with previous studies in the Results and discussion section.

15.  Also, the author should separate the different topics into the same paragraphs. Currently, the same paragraph deals with various topics in the Results and discussion section.

16.  I suggest merging the sub-section of “Policy implications” with Conclusion section.

17.  I recommend adding some limitations of the research to the Conclusion section as it would be useful to any reader.

18.  The English grammar and style should be checked throughout the manuscript (especially narrative format).

Reviewer 2 Report

The article is potentially interesting. But practically every sentence needs some correction to the English.

The emphasis on the role of government is interesting, particularly as in some other countries (developed) government supports but does not itself provide real extension services.

In relation to crop insurance programs, I am surprised there is no mention on maladaptation which has happened frequently with simple crop insurance programs. There is one exception where the government determined that it is best for farmers to be supported by farmer groups and a consultant and the results have been exceptional. Similarly in one jurisdiction, the agency in charge of a crop insurance program also provides advice to farmers who appreciate this government agency while they do not generally appreciate government itself!

Author Response

Point 1: The article is potentially interesting. But practically every sentence needs some correction to the English.

Response 1: Thank you for the positive response. The manuscript has been checked by MDPI English Editing services. We believe the current English is improved and suitable for publication.

Point 2: The emphasis on the role of government is interesting, particularly as in some other countries (developed) government supports but does not itself provide real extension services.

Response 2: Thank you for the comment. Actually, extension plays an important role in Indonesian rice farming. The government is the primary provider of extension services to rice farmer in Indonesia.

Point 3: In relation to crop insurance programs, I am surprised there is no mention on maladaptation which has happened frequently with simple crop insurance programs. There is one exception where the government determined that it is best for farmers to be supported by farmer groups and a consultant and the results have been exceptional. Similarly in one jurisdiction, the agency in charge of a crop insurance program also provides advice to farmers who appreciate this government agency while they do not generally appreciate government itself!

Response 3: We have difficulty at getting the essence of this comment. Can you provide an explanation on this comment? 

However, in the revised manuscript, we have incorporated the maladaptation issue. We believe that the revised manuscript has discussed this issue in detail.

Reviewer 3 Report

Major comments:

The quality of English is poor. It definitely requires a profesional proof reader. Otherwise it shouldn't be published. Grammatical errors are numerous. I suggest first to work on the content, then let a professional native speaker go through the article.  

Introduction and literature review

It's a great interesting subject, very relevant, it is abour climate risk perception as a driver of autonomous adaptation which is reflected well. However, sometimes the argumentation is too narrow, maybe also due to poor handling of the English language, so the main message  delivered seems to be that climate risk perception is THE driver of adaptation. In fact, it is one driver but there are also others: adaptation is also influenced by investment and running costs of adaptation strategy, trust in public institutions to cover the negative impacts of CC, trust in insurance and networks. This broasder picture is not provided.

A critical reflection is lacking. For example does high risk perception leads automatically to sustainable adaptation? Or can autonomus adaptation also be maladaptation?   

chapter 3 could be part of methods...

Methods:

rename the chapter

 to Methods with 4.1 data base, measurement

The variables (explanatory and dependent) should be described here, 

why the impact severity  was recoded as binary variables for analysis? not yet explained

which are the assumptions of ordered probit model - if any please mention it

isn't it necessary to control for some variables? region for example?

Results:

chapter starts with a section, that sounds like part of an abstract

The paper is lacking a systematic analysis/summary on which are the weakening and reinforcing factors of climate risk perception.

5.6 policy implications needs to be well structured. The region is mentioned here as factor,too.

maybe the significant factors can be shown in a graph such as:

gender : male ++++>  

                                                    climate risk perception

               female----->

etc

weakening ---------->

reinforcing ++++++>

more critically discussed, maladaptation? more chemical fertilizer? why land tenure is picked as recommendation for policy action (and extension and farmer-to-farmer extension); you have evidence for this selection?

Conclusion:
needs to be rewritten not much difference between actual and perceived impact... it is not so clear where you take this from? from part 5.5 its not yet evident

Minor comments:

219 Fig 1 : what do you mean by distribution of sample, where are the farmers with highest share of high impact?

223 Table 1. numbers for farmer field school missing, male and female should be male-headed household versus female-headed household, right?

234: male farmers: isn't male-headed rice farming household? or did you interview male and female,,,

264: Table 2 : provide also standard deviation

Round 2

Reviewer 1 Report

No further comments. 

Author Response

The reviewer does not provide any further comment.